# Assessing the Current Integration of Multiple Personalised Wearable Sensors for Environment and Health Monitoring

**DOI:** 10.3390/s21227693

**Published:** 2021-11-19

**Authors:** Zhaoxi Zhang, Prince Michael Amegbor, Clive Eric Sabel

**Affiliations:** 1Department of Environmental Science, Environmental Social Science and Geography, Aarhus University, 8000 Aarhus, Denmark; pma@envs.au.dk (P.M.A.); cs@envs.au.dk (C.E.S.); 2BERTHA, The Danish Big Data Centre for Environment and Health, Aarhus University, 8000 Aarhus, Denmark

**Keywords:** human sensors, individual data, physiological response, environment measurement

## Abstract

The ever-growing development of sensor technology brings new opportunities to investigate impacts of the outdoor environment on human health at the individual level. However, there is limited literature on the use of multiple personalized sensors in urban environments. This review paper focuses on examining how multiple personalized sensors have been integrated to enhance the monitoring of co-exposures and health effects in the city. Following PRISMA guidelines, two reviewers screened 4898 studies from Scopus, Web of Science, ProQuest, Embase, and PubMed databases published from January 2010 to April 2021. In this case, 39 articles met the eligibility criteria. The review begins by examining the characteristics of the reviewed papers to assess the current situation of integrating multiple sensors for health and environment monitoring. Two main challenges were identified from the quality assessment: choosing sensors and integrating data. Lastly, we propose a checklist with feasible measures to improve the integration of multiple sensors for future studies.

## 1. Introduction

The World Health Organization (WHO, 2018) recommends collective efforts to combat environment-related disease given the evidence on the effect of the environment on health and wellbeing. Research shows that stressors from the urban environment [1] are related to humans’ mental and psychological states whilst conducting outdoor activities in their daily lives. For instance, urban airborne particulate matter [2] poses serious health risks such as lung cancer and asthma, and road traffic noise [3] is associated with sleep deprivation and poor mental health. In addition urban form [4] effects our ability to do physical activity, leading to risk of being overweight, and all these health outcomes can vary by ethnicity and socio-economic status. In view of this, an in-depth investigation is advocated to assess the health effects of outdoor stressors in the urban environment.

With the advancement of sensing and wearable technology, personalized wearable sensors are now becoming ubiquitous. Coming from the Internet of Things (IoT) and quantified self (QS) (coined by Gary Wolf and Kevin Kelly in 2007, the term embodies self-knowledge through self-tracking) paradigms, personalized wearable sensors are now regularly used to track one’s own biological, physical, and behavioral [5] information, including psychological [6] and mental states [7] and physical activities [8]. Especially for outdoor activities, wearable sensors have the distinct advantages of portability and usability, enabling tracking people’s states during daily activity in the city [9,10]. Meanwhile, low-cost sensors for environmental exposure monitoring, carried by individuals, also benefit the self-tracking of personal exposures to specific outdoor stressors, such as PM_2.5_ (particulate matter with an aerodynamic diameter <2.5 μm) [11], noise [12] and radiation [13] in the urban environment. In light of these developments, employing personalized sensors can become an effective tool to monitor human’s outdoor physical activities and the health effect of outdoor stressors on humans.

The latest personalized devices such as the *SENSg* [14], are equipped with multiple sensors enabling the simultaneous monitoring of environmental exposures and human activity in the city. However, for the majority of studies, a more practicable and feasible method is to integrate multiple personalized sensors as a package to assess human health and environment during outdoor physical activity. 

However, research articles and reviews to date are limited to discussing the employment of sensors for environment monitoring [15] or health monitoring [16,17], but rarely to discuss the integrations between different kinds of sensors. Compared with employing one sensor, integrating multiple sensors brings more practical questions, such as the interoperability of multiple data from sensors [18], which have not been fully addressed yet in the literature. This significant knowledge gap regarding the implications of integrating multiple sensors makes it challenging for future research.

In view of the gap in the literature, this systematic review aims to investigate the application of integrating multiple sensors for health and outdoor environment monitoring in the city. Specifically, the review will: Assess current applications integrating multiple sensors for health and outdoor environment monitoring,Examine the main challenges related to the integration, andPropose workable approaches to optimize the integration and improve the feasibility of integration for future studies.

By reviewing current sensor-driven case studies, we hope to provide a framework upon which future studies can be based. 

## 2. Methods

### 2.1. Search Strategy

We reviewed papers published between January 2010 and April 2021. The rapid development of sensing technology significantly promoted the application of “personalised sensor” in recent years hence the rationale for using 2010 as the starting year. While studies prior to 2010 may use the words “monitor” or “device” in reference to “sensors”, they do not generally refer to personalized or mobile or miniature sensors. Our review adhered to the recommendations of the PRISMA (Preferred Reporting Items for Systematic Reviews and Meta-Analyses). Figure 1 shows the PRISMA flow chart of inclusion and exclusion of articles in the study’s identification steps. In this study, searches were conducted in five databases, namely: Scopus (244), Web of Science (3389), ProQuest (507), Embase (103), and PubMed (1570). Four categories of keywords and their combinations were used in the search: (1) ‘physical activity’ OR ‘outdoor activity’ OR ‘walking’ OR ‘cycling’; (2) ‘physical health’ OR ‘mental health’ OR ‘wellbeing’ OR ‘emotion’ OR ‘psychology’ OR ‘exposure’; (3) ‘environment’ OR ‘place’ OR ‘space’ OR ‘spatial’; (4) ‘wearable sensors’ OR ‘personal sensors’ OR ‘human sensors’. The combinations of searching results were “(#2 AND #3 AND #4) OR (#1 AND #3 AND #4)”. This review selected articles focusing on the integration of personalized sensors on monitoring environmental impacts on human’s responses during outdoor activities. Considering language barriers, only articles in English were included in the literature review. In addition, the reference list of relevant articles was reviewed to identify potential sources missed in the database search.

### 2.2. Selection Criteria

The search results were imported into ‘COVIDENCE’ (https://www.covidence.org (accessed on 20 August 2021)) for further assessment. After removing duplicates, the assessment in COVIDENCE left 4898 documents for the title and abstract screening. Two reviewers (ZZ and PMA) conducted the titles and abstract screening based on the predefined study inclusion criteria select papers, independently to avoid potential bias. The abstract and title screening resulted in 378 articles that met the inclusion criteria. The same reviewers conducted an independent full-text screening for eligibility. For the full-text review, studies were considered eligible for our review if they: (1) related to environmental impacts and health outcome; (2) were conducted in a real-world setting and an urban environment; (3) collected data via sensing technology; and (4) employed multiple personalized sensors (at least two different sources of personalized data). We excluded reviews and reports. Disparities in the assessment were resolved by consensus discussion between the two reviewers.

### 2.3. Data Extraction

For each study, the following basic study characteristics were extracted (Table 1): the year of publication; temporal resolution (e.g., hours, days, months, seasons); study setting (natural and predefined settings); area of publication (e.g., social science, environmental science, geography, medical research); region of study (e.g., Asia, Europe, North America); participants’ gender and sample size; types of environment (e.g., nature, social environment, built environment and physical environment); types of health (behavior, physiological health); other contextual (Yes/No) and geospatial data (Yes/No). In addition, information about data collection, such as measures for the environment and health-related outcome, sensor packages, statistical analysis and Appendix A, were extracted (Appendix A) to provide a summary of the multiple sensors used in the study of environment and health.

### 2.4. Quality Assessment

To assess the methodological quality of reviewed studies, we used the assessment checklists adapted from the assessment tool Evaluation of Public Health Practice Projects (EPHPP) [19] and Guidelines for Critical Review of Qualitative Studies [20]. It has been shown that EPHPP is an effective tool to systematically assess the quality of quantitative studies [21]. The tool has previously been used in systematic reviews on the environmental effects on human wellbeing [22,23] and physical activity [24,25]. EPHPP has seven categories: selection bias, study design, confounders, blinding, data collection method, withdrawals and analysis. 

Concerning that the integration of multiple personalized sensors is a new and interdisciplinary method, we hope to assess the completeness and systematics of its applications. Hence, we added criteria covering the qualitative research from the Guidelines for Critical Review which are not covered by the EPHPP, including study purpose, literature review, and conclusion. In reference to the adaption used by Won, et al. [26], the checklist (Appendix A) was applied in this paper on the basis of assessment criteria for study purpose, literature, sampling (description, representation, consent), study design, data collection method (description and tool), withdrawals, confounders, data analysis and conclusions. 

In the checklist, ¨0¨ means ¨Weak¨ (“no”), ¨1¨ means ¨Moderate¨ (“yes”), ¨2¨ means ¨Strong¨, the total score range from 0 to 20. The papers scores ranges from 8 to 19 with eight studies (20.5%) with scores of 8–11 in the low-quality category, 21 studies (53.8%) with scores of 12–15 in the middle-quality category and 10 studies (25.6%) reached the high-quality category with scores of 16–19. The result of the assessment is provided in Appendix A.

**Table 1 sensors-21-07693-t001:** Characteristics of the reviewed studies.

NO	RE	Temporal Resolution	Subject Area ^1^	Location	Study Setting	Gender	Sample Size (Include) (Age Group)	Environment Type	Geo-Data	Contextual Data	Health
1	Benita, et al. [27]	10 min	Social science, Environmental Science	Singapore	Pre-defined (700 m walking)	Female	10 (aged 21–25)	Physical	Yes	Yes	Activity and Mental health
2	Benita and Tunçer [28]	10 min	Environmental Science, Agricultural and Biological Sciences	Singapore	Pre-defined (700 m walking)	Female	10 (aged 21–25)	Physical and Urban	Yes	Yes	Activity and Mental health
3	Birenboim, et al. [29]	30 min	Social science, Earth and Planetary Sciences	Netherlands	Pre-defined (3 km walking)	Male	15 (12) (average age of 21.8)	Urban	Yes	Yes	Mental health
4	Bohmer, et al. [30]	7–10 days	Arts and humanities, Medicine, Neuroscience	Netherlands	Natural	Both	82 (48) (average age of 62.3)	Physical	No	No	Activity
5	Boissy, et al. [31]	14 days	Medicine	Canada	Natural	Both	75 (54) (aged 55–85)	Urban	Yes	No	Activity
6	Bolliger, et al. [32]	15 days	Environmental Science, Medicine	Belgium	Natural	Both	5 (Adults)	Social	Yes	No	Mental health and Psychology
7	Borghi, et al. [33]	14 days (repeat in two seasons)	Environmental Science, Medicine	Italy	Pre-defined (90 km home-to-work)	-	1 (Adult)	Physical	Yes	No	Physical health
8	Burgi, et al. [34]	7 days	Multidisciplinary	Switzerland	Natural	Both	123 (119) (aged 11–14)	Urban	Yes	Yes	Activity
9	Butt, et al. [35]	14 days	Medicine	USA	Natural	Both	20 (11) (aged 24–35)	Social	No	No	Activity
10	Cerin, et al. [36]	7 days	Medicine, Health Professions	USA	Natural	Both	84 (73/66) (aged 3–5 children and their parents)	Urban	Yes	No	Activity
11	Chaix, et al. [37]	7 days	Medicine, Health Professions	France	Natural	Both	319 (285) (average age of 50.2)	Urban	Yes	No	Activity
12	Chrisinger and King [38]	20–25 mins	Medicine, Computer Science	USA	Pre-defined (One walk route)	Both	14 (Adults)	Social and urban	Yes	No	Mental health
13	Dessimond, et al. [39]	6.5/8 days	Engineering, Medicine, Computer Science	France	Natural	-	1 (Adult)	Physical	Yes	Yes	Activity
14	Do, et al. [40]	7 days	Environmental Science, Engineering, Earth and Planetary Sciences	USA	Natural	Both	18 (Adults)	Physical	Yes	Yes	Activity
15	Doherty and Oh [41]	3 days	Medicine, Health Professions	Canada	Natural	Both	40 (37) (aged 32–75)	Urban	Yes	No	Physical health and Activity
16	Donaire-Gonzalez, et al. [42]	1 day (repeat in three seasons)	Environmental Science	Europe (Five cities)	Natural	Both	158 (average age of 61)	Physical	Yes	Yes	Activity
17	Doryab, et al. [43]	16 weeks	Medicine	USA	Natural	Both	188 (160) (college student)	Social	Yes	No	Activity
18	El Aarbaoui and Chaix [44]	7 days	Environmental Science, Medicine	France	Natural	Both	78 (75) (aged 34–74)	Physical	Yes	No	Physical health and Activity
19	Engelniederhammer, et al. [45]	Around Midday	Social science	China	Pre-defined (walk route with 4 street paths)	Both	30 (average age of 24.77)	Social	Yes	No	Mental health and Psychology
20	Huck, et al. [46]	days	Environmental Science, Medicine	UK	Natural (different routes)	Male	1	Physical	Yes	No	Physical health
21	Johnston, et al. [47]	18 h	Environmental Science, Medicine	USA	Natural	Both	18 (10) (aged 15–17)	Physical	Yes	No	Psychology
22	Kanjo, et al. [48]	45 min	Computer Science	UK	Pre-defined (shopping route)	Female	40 (average age of 28)	Physical	Yes	No	Mental health and Psychology
23	Kim, et al. [49]	Hours	Social science, Environmental Science	USA	Pre-defined (1.26 km walking route)	Both	30 (average age of 24.2)	Urban	Yes	No	Physical health and Activity
24	Kou, et al. [50]	A weekday and a weekend day	Social science, environmental Science, Engineering	USA	Natural	Both	46 (33) (18–65)	Physical	Yes	No	Activity
25	Laeremans, et al. [51]	7 days (three times in different seasons)	Environmental Science	Europe (three cities)	Natural	Both	122 (average age of 35)	Physical	No	No	Physical health and Activity
26	Ma, et al. [52]	A weekday and a weekend day	Environmental Science	China	Natural	Both	177 (97) (aged 18–60)	Physical	Yes	No	Activity
27	Ma, et al. [53]	A weekday and a weekend day	Social science, Earth and Planetary Sciences	China	Natural	Both	177 (112) (aged 18–60)	Physical	Yes	Yes	Activity
28	Millar, et al. [54]	Hours	Social science, Environmental Science	Netherlands	Pre-defined (18 km long between urban and rural)	Both	12 (half aged 18–24, the remaining half were older 55)	Urban	Yes	Yes	Mental health
29	Novak, et al. [55]	7 days	Engineering, Medicine, Computer Science	Slovenia	Natural	Both	2 (Adult)	Physical	No	No	Physical health
30	Ojha, et al. [56]	Hours	Engineering, Computer Science	Switzerland	Pre-defined (1.3 km walking)	-	30 (-)	Physical and Urban	Yes	Yes	Mental health
31	Rabinovitch, et al. [57]	4 days (twice in two non-consecutive weeks)	Medicine	USA	Natural	-	30 (schoolchildren average age of 10)	Physical	Yes	No	Physical health
32	Resch, et al. [58]	Hours	Environmental Science, Medicine	Europe (two cities)	Natural	Both	56 (over 18)	Urban	Yes	No	Mental health and Psychology
33	Roe, et al. [59]	Unassisted walking for 15–20 min	Medicine	USA	Pre-defined (two routes: “green” and “gray”)	Both	11 (aged 65)	Physical	Yes	No	Physical Activity and Mental health and Psychology
34	Runkle, et al. [60]	5 days	Environmental Science	USA (three sites)	Natural	Both	66 (35) (Average age around 38/39)	Physical	Yes	No	Physical health
35	Rybarczyk, et al. [61]	Hours	Social science, Engineering	Germany	Natural (within 1.1 km^2^)	Both	28 (aged 20–70)	Urban	Yes	Yes	Physical health and Activity
36	Shoval, et al. [62]	1 day	Social science	Israel	Natural	Both	144 (68) (aged over 18)	Urban	Yes	No	Mental health and Psychology
37	Steinle, et al. [63]	days, Repeat in winter and summer	Environmental Science	Scotland	Natural	-	17 (-)	Physical	Yes	Yes	Activity
38	West, et al. [64]	14 days	Social science, Environmental Science	Kenya	Natural	Both	6 (aged 18–55)	Physical	Yes	Yes	Psychology
39	Zhang, et al. [65]	A weekday and a weekend day	Medicine, Computer Science	China	Natural	Both	156 (138) (aged over 18)	Physical and social	Yes	No	Psychology

^1^ The subject area of publication can be found on the website of SCImago Journal and Country Rank (https://www.scimagojr.com/ (accessed on 8 November 2021)), which is a public platform to assess and analyze scientific domains of journal.

## 3. Results

### 3.1. Assessment of Current Application

To assess the current development of applying multiple sensors, we present the characteristics of the reviewed studies in Table 2. Of the 39 studies, more than 95% of studies were published after 2015, with 41.0% of studies were published after 2020, indicating that the application of multiple sensors is a rapidly evolving research topic. Most of our reviewed papers were in Europe (46.2%) and North America (33.3%), which may be attributed to limiting the publication language to English. Nonetheless, the number of papers in Asia has increased in the most recent years (3 of 16 from 2020 to 2021 April). In the reviewed studies, the application of multiple sensors occurred mainly in single cities. Only 10.3% of them compared their application between cities (three in Europe and one in the USA).

The majority of applications (74.4%) recruited male and female participants, and 46.2% of them recruited 10–50 participants, while a small number of studies focused on children and elderly groups of people [57,59]. 71.8% of the studies applied multiple sensors in a natural setting, while 28.2% chose to collect data in pre-defined and controlled settings. For example, in one study, participants wore sensors and walked along a 3 km path without talking [29]. 33.3% of studies employed multiple sensors merely for several hours within a day, 53.8% measured for over one day and up to weeks, and only 12.8% repeated the experiment in different months during a year. 

Although integrating multiple sensors for health and environment monitoring is increasingly used these days, it is still at the early stage of development. First, the current applications are not applicable to all age groups and the small samples make it hard to generalize the results for the wider population. Second, a few studies repeated the tracking for a longer time, but we are still far away from monitoring environment-related chronic disease through a life course [66]. Lastly, the integration is not integrated enough to track in a natural manner in various urban settings, and the results are not comparable between different cities. Therefore, there is a lot of room to improve integration between sensors.

### 3.2. Two Challenges for Integration

The review examined the grade of papers—based on the quality assessment checklist—to identify the current challenges that might limit the quality of the reviewed studies. For the low-quality category of papers, all of the 8 studies that performed weakly on the criterion of representative sampling due to small sample size or selection bias, also had a weak performance on the data analysis. The primary challenge identified for reviewed studies in the middle-quality category (21 studies) was the study design. For example, over 50% (12 of 21 studies) only employed the sensors for a short period (e.g., half hour, 45 min) or short distance (e.g., 3 km), which may “miss” hidden problems, such as people’s tolerance of sensors, and not be enough to test the integration for long-distance tracking. Around 70% of them did not acquire a high score on data analysis, indicating the prevalence of technical weakness in analyzing the data. 

The aforementioned challenges show that recruitment and measurement are essential to the research quality, however, they depend on the sensors used. That is to say, the integration may skew the result of recruitment in practice. In addition, in contrast with traditional data, it is essential to fuse data from different sensors and extract the features before conducting any advanced analysis. The potential use of data may be limited without proper data fusion. 

Therefore, two unsolved challenges were found in the reviewed papers: Sensors and sampling: how to choose and integrate sensors reasonably and form a workable integration in fieldwork to solve the research questions effectively; andData fusion and database: what are the techniques required to link up data and build up a high-quality database for the subsequent analysis.

To cope with the challenges, the next section of this review focuses on issues related to sensor integration and data fusion (Table 3). In the following analysis, we illustrate the considerations of sensors and optional data fusion techniques that may lead to the success of integration. Lastly, we will summarize the knowledge learnt from the 39 reviewed papers and recommend a new approach to cope with these challenges.

### 3.3. Challenge 1: Sensors and Sampling

#### 3.3.1. The Form of Integration

Nearly 50% of the reviewed studies integrated portable environment monitors and health trackers (“Environmental monitors”+ “health trackers” + …) in order to measure the health effects triggered by specific environmental factors. Most of them (89.7%) added location tracking sensors (e.g., GPS receiver, phone-based GPS, sensor-based GPS) to capture the geo-information. Additionally, 25.6% combined GPS with portable environmental monitors (“GPS” + “Environmental monitors”) to map the spatial features of environmental stressors and around 7.7% combined GPS with health trackers (“GPS” + ”health trackers”) to study the spatial relationship between geography and physiological effects. As an accelerometer can examine the speed, direction and acceleration of a user, around 10.3 % of reviewed studies combined accelerometer and GPS to record the trajectories of physical activity, and 25.6% integrated GPS and accelerometers with environment monitor/health trackers (“GPS” + “Accelerometer” + …) to measure the activity-centric exposures or lifestyle-related diseases. 

#### 3.3.2. Number of Sensors

Is using as many as possible sensors be good for research? 53.8% of reviewed papers employed two sensors, and 30.8% integrated three kinds of sensors as a package, while 12.8% employed four sensors and only 2.6% employed five sensors. A sensor package consisting of five sensors might be challenging for long-distance walking [56] and might increase the weight for carrying, which could lead to onerous experiences and fatigue. Research papers emphasized that sensors should be light and small, and not be burdensome for participants [63]. Therefore, it is fundamental to build a simply and light integration of sensors. Additionally, with increasing numbers of sensors, data fusion becomes more challenging.

#### 3.3.3. The Cost-Effectiveness of Sensors

The cost-effectiveness of sensors relates to the cost, function, accuracy and applicability, which is crucial to the decision-making process of choosing sensors. Appendix A shows more than 30 sensors from the 39 reviewed studies. Here, the aim is not to analyze each sensor’s usability, but to generalize the considerations of every category of sensors. 

Foremost, GPS data is widely used to locate sensors. 35.9% of studies employed GPS receivers, and the cost of a GPS receiver is around $70–$240. An accelerometer is a light and cheap sensor, enabling monitoring the level of physical activity (PA) in outdoor activities, and which costs around $35–$150. Additionally, 56.4% utilized smartphone-based/tablet-based/sensor-based GPS or accelerometers to lower the expense of sensors. As for the effectiveness, the reliability of phone/sensor-based GPS or accelerometer depends on the accuracy of the mobile device and the position of the smartphone device [67], and researchers stated that the GPS receiver might miss data when the GPS signals from satellites are weak [68]. In Donaire-Gonzalez, et al.’s [42] study, participants also carried a GPS receiver and an accelerometer to validate the data collected by phone application.

The most commonly applied health tracker in the integration is the wristband (46.2%). Compared with a lab-based medical instrument, wristbands are light and user-friendly, which are easy to use in normal life. The cost of a wristband varies from hundreds to thousands of dollars. For example, the medical-level wristband Empatica 4(E4) costs $1690 [38,49], while the fitness-level wristbands (e.g., Fitbit and Garmin) cost $150 to $400 [43,60], but E4 has higher resolution of data (f = 4 Hz). In addition, data from E4 is accessible in its raw form [62] and is visualized on an online cloud platform, while Microsoft Band 2 (MS Band) does not permit straightforward raw data exportation [29], thus a third-party application is required to log raw data. As for Fitbit, the data can be retrieved by the Fitbit app programming interface (API), but the data quality has not been reported [43]. Birenboim, et al. [29] compared the cost-effectiveness between Empatica and Microsoft band in their paper and suggested to test the devices in different environments for longer exposure time.

As for the sensors monitoring environments, the majority of integration was applied in the measurement of the physical environment: mainly 38.5% employed air pollutant sensors, 25.6% employed noise sensors, 20.5% employed temperature sensors. In contrast with health trackers, the price of portable environment monitors is flexible, since self-assembled environment monitors can decrease costs [39], but they are not as accessible as the health wristbands from the market. To test the accuracy, researchers often co-located sensors with fixed monitors in the city [53] or compared the data from sensors with the official environmental indexes [55].

Another workable way to lower the number, cost and weight of sensors is employing phone-based sensors and applications (43.6%), for example, phone-inbuilt temperature sensors [47], microphone to measure noise [28], and Bluetooth to measure the social distance [35]. In addition, phone-based application can assist the measurement in the field [41] and control data [46], such as “ExpoApp” [42], which is an integrated system to assess multiple personal environmental exposures.

### 3.4. Challenge 2: Data Fusion and Database

#### 3.4.1. Data Logging

Modern mobile technology encourages loading data from multiple sensors easily. In this case, 10 papers (25.6%) employed a phone application to control streaming data, which enables compressing, and data integration. Five papers (12.8%) could transmit data to a remote project server [41,46]. In these two situations, some of them can store data in the sensors if Wi-Fi connectivity was unavailable [40]. However, for most studies (66.7%), data was only stored locally in the sensors, leaving data to be exported via Universal Serial Bus (USB) to computer, then the following techniques are important to quality of integrating data.

#### 3.4.2. Pre-Processing

Pre-processing data aims to review the data, enabling: (1) extracting features (23.1%); (2) exclusion of malfunctioned, negative or zero values (20.5%); (3) classification of data or label data (17.9%); (4) the use of corrections or weights to the data (12.8%); (5) filtering noise (23.1%); (6) including data by the threshold (7.7%); (7) normalizing or standardizing data (12.8%); and (8) smoothing data (7.7%). In our reviewed papers, the pre-processing usually includes three to five steps to clean data and make it ready for analysis (Table 3). Since manually reviewing data requires a higher workload [34], 20.5% took advantage of specific software/toolbox and 28.2% used algorithms to observe and deal with the data, while 7.7% finished this procedure through a smartphone application. 

Sometimes, to remove noise caused by technical inaccuracies, advanced signal processing techniques are utilized. Usually, the Butterworth filter performed well on removing higher or lower frequency variations. For example, Resch, et al. [58] filtered Galvanic skin response (GSR) by a low-pass filter at 0.5 Hz and a high-pass filter at 0.05 Hz; Boissy, et al. [31] filtered accelerometer data by the low-pass filter at 5 Hz and high-pass filter at 1 Hz; and Kim, et al. [49] filtered accelerometer data by a low-pass filter with cut-off frequency at 4 Hz. The frequency response of a filter is dependent on the sensor/data frequency.

#### 3.4.3. Unification

Unifying the temporal components and frequency is crucial for fusing data, due to the fact that health trackers usually have higher resolution than other kinds of sensors (Table 3). There are two ways to process the sampling rate of a signal: (1) Time interpolation (12.8%), also called as “up sampling”; and (2) Frequency reduction (10.3%), also known as “down sampling”. 

Time interpolation, to increase the sampling rate of low-resolution data to the same level as high-resolution data, preserves the higher frequency. It can obtain higher resolution datasets [40] and enhance the precision in the analysis [54], but it may increase the computation as well. Otherwise, frequency reduction, to decrease the sampling rate of high-resolution data to the same level with low-resolution data, is often used to merge health data [29,62], accelerometer data [36] with GPS points, since it can save the calculation time of spatial analysis, but may lead to the loss of resolution to some degree.

For studies employing sensors with many variations of frequency, it is critical to decide the sampling rate for fusing data; 15.4% employed a moving average to decrease the possible loss of resolution. For example, in Benita and Tunçer [28], they use a moving average (f = 1 Hz) to link up temperature, wind and atmospheric pressure (f = 0.5 Hz), phone-based GPS and speed (f = 0.2 Hz), EDA data (f = 4 Hz) and noise (f = 10 Hz). In the reviewed papers, the window size for moving average varies from 1 min [44,64] to 5 min [44], which is dependent on the data features. For health data, we recognize that the window size should also follow the related health metric, since human responses have different latency time. For example, a 5-min window for measuring short-term Heart Rate Variability (HRV), and 1-min windows for assessing HRV dynamics that may be masked within 5-min windows [44]. In the process of finding the best window size, Ojha, et al. [56] observed that for EDA data time-windows vary from 0 to 12 sec, then standardized at 5-sec time-windows. 

Therefore, the decision-making around unified frequency and window size for moving averages is a critical process. It is essential to take the considerations of research objectives, variables, the capacity of calculation and related knowledge of regulations into consideration.

#### 3.4.4. Data Aggregation

There are three main forms of aggregation: (1) Aggregating the data (15.4%) over time to compare changes within a period of time (e.g., from morning to night, weekdays and weekend); (2) Aggregation by multiple individuals (7.7%) to indicate the difference between individuals, genders, age groups and socioeconomic factors [62]; and (3) Aggregation by spatial units (25.6%) based on geographical information. Three papers used three aggregations at the same time to describe data from the perspective of “time”, “individuals” and “location” [63,64]. For deeper analysis, 12.8% of studies aggregated the data over time and space to explore the spatial-temporal features of human activities on site, 10.3% of them combined spatial analysis and socioeconomic analysis to explore the interrelationship between society and geography. 

Benefitting from GPS data, over 50% of reviewed papers aggregated data over spatial units, such as based by 20 m × 20 m cellular network [62], 50 m grid squares [64], microenvironment types [63] and specific Locations with 50 m/100 m buffers [36]. Having spatial data, some papers utilized spatial clustering algorithms such as Hotspot analysis [27,28], Getis-Ord Gi* local statistics and kernel density [38,58] to understand spatial features of environmental effects in Geography Information System (GIS).

### 3.5. How to Improve the Integration

The integration of multiple sensors includes two processes: combining different sensors smartly and effectively integrating data from different sensors. Compared with previous sensor research, although integrating multiple personalized wearable sensors is dependent on the tracking, this method requires more critical decision-making regarding the costs, recruitment of participation, implementation and validity of data. By critically reviewing 39 papers, this paper identifies a checklist with crucial issues: preparation, sensor selection, data collection, data integration and analysis (Table 4) to improve the feasibility of integration in the future. The general recommendations made in Table 4 are necessarily generic guidelines. For specific health outcomes or environmental exposures, modifications to these guidelines would be expected.

## 4. Discussion and Conclusions

### 4.1. Discussion

The integration of multiple personalized wearable sensors offer an opportunity to contribute to environment and health research, but it is still at the preliminary stage. The weaknesses of integrating multiple personalized wearable sensors cannot be ignored either. First, many studies [35,41,43] admitted that bias exists in sampling (e.g., approach and size). For example, Butt, et al. [35] emphasized that their subjects tended to be healthy, well-educated young adults, predominantly married mixed-gender couples, while Doryab, et al. [43] purposely chose university students. The sub-discipline of using wearable sensors might thus still be in its infancy, with results not yet generalizable to a broader population.

Cost remains a significant factor in the success of a campaign. To improve the cost-effectiveness, issues related to the weight of the sensor packages [42], cost and accessibility [37], operability [41], accuracy [54], comfort or ease of use [49] and the accessibility of raw data [62] should be taken into considerations. That may explain why studies that employ various kinds of environmental sensors and health trackers only recruit a small number of volunteers, since the cost, weight and number of sensors can skew the small sample size.

Many studies conducted their research in a single city, but the local weather and season [34,49], geography [42] and culture [45] could make it challenging to switch locations and completely duplicate the experimental settings [44]. Participants are also limited to a predefined environment or path in different studies, thus the results are non-comparable between different environmental settings. 

Longitudinal tracking provides the ability for in-depth investigations, but so far, this has rarely been done. Current experiments mainly focus on specific groups, such as Type 2 diabetic patients [41], teenagers [14], and children and parents [34,36]. In addition, some low-cost sensors for environmental monitoring are not applicable in longitudinal tracking due to the weight and size of sensors. 

Based on the above discussion, the integration of sensors is a process of critical decision-making. The employment of sensors is not only associated with the costs and functions but also tightly connected with the participants (age, gender, education, sample size), their willingness, and careful thinking of whether the sensors are user-friendly, whether the sensors enable longitudinal tracking, how long the participants need to wear and whether it may cause risks for participants. We summarized all the factors related to the preparation and the integration of sensors in the checklist (Table 4), hoping to inspire more studies to complete this process.

In the field, some studies attributed missing data to unavailable or weak signals, non-response of sensors, loss of battery power and the failure of the application while using a phone [56]. Since the real-world situation is complex and filled with interference, it is necessary to consider the data quality in fieldwork, such as battery expiration, signal loss, data loss, and data noise [30,64,65]. 

As shown in Table 4, before implementation, it is essential to test the reliability of sensors, including to check, charge, calibrate and set up the sensors prior to use [14,43]. It is also recommended to use an extra sensor as validation [42] or to co-locate the mobile data from a fixed monitor station [55]. Lastly, it is essential to have pre-training to teach participants the operation of sensors [14].

Smartphone applications show a great potential to log, store data, integrate and visualize data automatically [41]. Some commercial companies such as Empatica and Fitbit also provide accessible APIs (Application Programming Interface) to connect sensors with other applications [43]. Software developers can develop applications, enabling the monitoring of numerous sensors to assure continuous functionality in the field [41]. 

To extract the data features, professional software is useful, but often needs a paid license, which increase the cost. Further, advanced methodological and analytical techniques, such as machine learning [43], can be utilized to deal with the ever-growing data from multiple-sources, but it might demand high computing power [54], which also requires further financial support. 

To compare the data from different sensors, it is helpful to comply with standardized terminology and nomenclature according to the official regulations [55]. For example, the duration of sleep in our reviews was assessed by “total sleep time (TST)”, “the number of wakenings”, and “time taken to fall asleep” by medical devices [35], and by the average length of “asleep” or “awake” by Fitbit [43]. The inconsistency of terminology may increase the difficulties to repeat the experiment with different sensors, thus we recommend describing the features of sensors and data by unified criteria. For example, the format of sensor frequency is not uniform (Table 3) in reviewed papers; there are many descriptions, such as “1 HZ”, “every 1 s”, “1 recording 1 s”, “10 readings per 1 s”, ”1-min epochs”.

Recent advancements in information technology have led to the emergence of various wearable products such as clothes, belts, watches, wristbands and cameras [69]. An integrated system with multiple sensing functions will promote the development of IoT (Internet of Things), and the tendency of integrating a variety of sensors is irresistible. However, there is still a long distance from now to the future. The factors related to integration discussed in this review are crucial to the next exploration. 

### 4.2. Strengths and Limitations of Our Review

Integrating multiple personal wearable sensors enables an individual-centered research paradigm, but this is still an embryonic field with few research outputs. In view of this shortfall, this review paper is the first to assess the feasibility of integrating multiple sensors for monitoring environment and health. The overarching contribution of this paper is reviewing the above knowledge in integrating multiple sensors and suggesting a checklist to improve feasibility and overcome the deficiencies.

The main limitation is that we were unable to rank the performance of each sensor package, since there was a huge variety of devices amongst the papers we reviewed, and these papers were not consistent in reporting performance. Another limitation is that this paper did not address the combination of objective measurement and subjective survey (e.g., mobile-survey, questionnaire), since this review mainly centers on the integration of different sensors and data. Future review studies might discuss how to combine sensors with qualitative surveys to understand human motivations, preferences and experiences.

### 4.3. Conclusions

In conclusion, this review assessed and characterized the state-of-the-art in integration of multiple personal sensor packages for outdoor environments, and summarized improvements needed in the future. Integration of personalized wearable sensors can enhance the ability to reveal relationships between environmental context and health outcome(s). Lastly, it is hoped that the rigorous methodology demonstrated in this review paper will provide a framework to enhance the ability of future studies to address further challenges in investigating the complex relationships between natural and social environments and human health, using multiple, personal sensors.

## Figures and Tables

**Figure 1 sensors-21-07693-f001:**
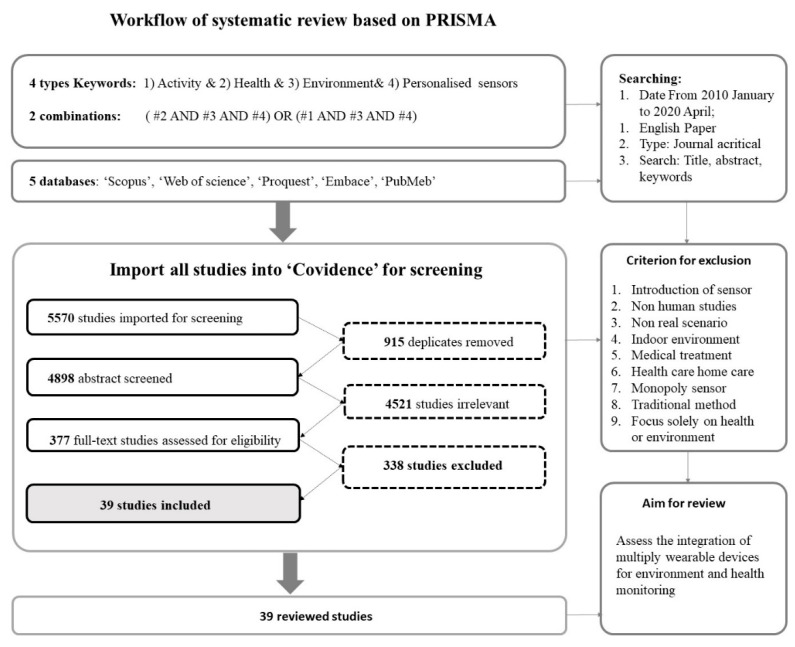
Article selection process.

**Table 2 sensors-21-07693-t002:** Characteristics of included studies (k = 39).

Study Characteristics	No.	%
**Publication year**		
2010–2015	1	2.6%
2015–2020	22	56.4%
2020–2021 April	16	41.0%
**Temporal resolution**		
Level 1 (Minutes/Hours within a day)	13	33.3%
Level 2 (Days/Weeks)	21	53.8%
Level 3 (Months and Seasons)	5	12.8%
**Subject area**		
Social science	9	23.1%
Environmental science	18	46.2%
Engineering	6	15.4%
Arts and humanities	1	2.6%
Medicine	19	48.7%
Computer Science	6	15.4%
Multidisciplinary	1	2.6%
Earth and Planetary Sciences	3	7.7%
Neuroscience	1	2.6%
Health Professions	3	7.7%
Agricultural and Biological Sciences	1	2.6%
**Region of study**		
Asia	6	15.4%
Europe	18	46.2%
North America	13	33.3%
Other	2	5.1%
**Locations**		
Single area/city	35	89.7%
Two or more areas/cities	4	10.3%
**Study setting**		
Natural settings	28	71.8%
Pre-defined settings	11	28.2%
**Gender**		
Both male and female	29	74.4%
Female only	3	7.7%
Male only	2	5.1%
Not mentioned	5	12.8%
**Sample size**		
<10	6	15.4%
10–49	18	46.2%
50–100	6	15.4%
>100	9	23.1%
**Domains of environment**		
Social environment (such as crowdedness, sociality)	4	10.3%
Urban environment (such as built environment, traffic)	11	28.2%
Physical environment (such as noise, air, wind, light)	20	51.3%
Physical and urban environment	2	5.1%
Social and urban environment	1	2.6%
Physical and social environment	1	2.6%
**Geo-location data**		
Yes	35	89.7%
No	4	10.3%
**Other contextual data**		
Yes	13	33.3%
No	26	66.7%
**Domains of health**		
Human activity (such as, physical activity, sleep)	14	35.9%
Physical health (such as, health condition, disease)	5	12.8%
Mental health (such as, stress)	4	10.3%
Psychology	3	7.7%
Human activity and Mental health	2	5.1%
Human activity and physical health	5	12.8%
Human activity and psychology	0	0.0%
Mental health and psychology	5	12.8%
Mental health and physical health and psychology	1	2.6%

**Table 3 sensors-21-07693-t003:** Characteristics of the reviewed studies.

	Reference		Integrate Sensors					Integrate Data from Sensors			
	RE	Q ^1^	N ^2^	Environment Sensor	GPS	Activity	Health Tracker	Data Logging	Pre-Processing	Data Fusion	Aggregation
1	Benita, et al. [27]	M	3	(1) Kestrel 5400 : temperature, relative humidity, wind and atmospheric pressure (one reading every 2 s); (2) Phone: Noise (10 readings per second).	(2) Phone-based GPS and speed (one reading every 4 or 5 s)	-	(3) Empatica 4 (four readings per second)	Sensors	(a) Filter the noise; (b) Extract feature by Ledalab software ^3^.	A moving average to smooth data (f = 1 Hz)	Over spatial units (stress hotspot)
2	Benita and Tunçer [28]	M	3	(1) Kestrel 5400 : temperature, relative humidity, wind and atmospheric pressure (f = 0.5 Hz); (2) Phone: Noise Noise (f = 10 Hz).	(2) Phone-based GPS and speed (f = 0.2 Hz)	-	(3) Empatica 4 (f = 4 Hz)	Sensors	(a) Filter the noise; (b) Extract feature by Ledalab software.	A moving average to smooth data (f = 1 Hz)	Over spatial units (stress hotspot)
3	Birenboim, et al. [29]	M	3 (App)	-	(1) GPS receiver (f = 1 Hz)	-	(2) Microsoft Band (f = 1 Hz); (3) Empatica 4 (f = 4 Hz).	Sensors/Phone App	(a) Extract feature by Ledalab Software; (b) Use t-test to detect significant differences between “neutral” and “stressful”.	Reduction (f = 1 Hz)	Over spatial units (Average per walking segment)
4	Bohmer, et al. [30]	H	2	(1) Light sensor (measured in 1-min epochs)	-	(2) Accelerometer (sum activity counts for 1-min epochs)	-	Sensors	(a) Transform lux to log lux; (b) Only include timeframe with <25% missing data; (c) Actant –Activity Analysis Toolbox to calculate the bedtimes; (d) Filter by thresholds of 50 min >1000 lux.	Average illuminance (log lux) per minute	Over Time (Average per timeframe)
5	Boissy, et al. [31]	H	2	-	(1) GPS receiver (-)	(2) Accelerometer (-)	-	Sensors	(a) Filter accelerometer data by low-pass filter at 5 Hz and high-pass filter at 1 Hz; (b) Use algorithm to detect step and remove noise; (c) Filter GPS points with lower presion.	Time interpolation; (Open-source software ^4^ to format data coming from the different sensors).	Over spatial units (clusters and transit detected by a rolling window).
6	Bolliger, et al. [32]	L	2 (App)	(1) Phone: Light sensor, temperature sensor and voice sensor.	(1) Phone based GPS	-	(2) Empatica 4 (f = 4 Hz) (App based ecological momentary assessment).	Sensors	-	-	-
7	Borghi, et al. [33]	L	4	(1) DiSCmini: UFP exposure levels; (2) A PM_2.5_ concentration monitor; (3) CairClip NO2; (All sensors: an acquisition rate equal to 60 s).	(4) Sensor-based GPS (Suunto 9)	-	(4) Heart rate monitor: Suunto 9	Sensors	Correct the particulate matter (PM) data by a correction factor; Exclude zero and unreliable data.	Average values over time	Over Time (season)
8	Burgi, et al. [34]	M	2	-	(1) GPS receiver (at 10 s intervals)	(2) Accelerometer (-)	-	Sensors	Manually reviewed	Software (Actilife 6.5.2, Actigraph, Pensacola, FL, USA)	Over spatial units (based on the activity settings) and Individuals (by gender).
9	Butt, et al. [35]	M	2 (App)	(1) A software platform on phone: (actual time a person spent interacting and the number of people with whom there were interactions).	-	-	(2) Wireless system (sleep, eye movement)	Sensors (Digital card);	(a) Calculate the median value of paramters; (b) Normalize value of social exposure between 0 and 1; (c) Performe Spearman’srank correlations to understand data.	-	Over Individuals (Wilcoxon sign-ranked test and 2D k-means clustering).
10	Cerin, et al. [36]	H	2	-	(1) GPS receiver (30 s epochs)	(2) Accelerometer (15 s epochs)	-	Sensors	(a) Remove periods of 30+ minutes of zero accelerometer counts; (b) Extract valid Accelerometer data (≥480 min of activity data/day); (c) Classify into sedentary time and MVPA ^5^ by cut points; (d) Use a web application (PALMS) ^6^ to clean and filter accelerometer and GPS data.	Reduction (Average accelerometer counts per 30 s)	Over spatial units (Average based on specific Locations with 50 m buffer/100 m buffer) and over individuals (average by gender and weight status).
11	Chaix, et al. [37]	M	2+1 ^7^	-	(1) GPS receiver (one point every 5 s)	(2) Accelerometer (-) (Phone mobility Survey in a web mapping application)	-	Sensors	(a) Use Web application (TripBuilder) to process GPS data; (b) Removed incorrect trips manually; (c) Use software (ActiLife 6.11.9) to process accelerometer data.	-	Over spatial units (calculate the percentage of the walk distance in main travel modes and test the differences by KruskalWallis test).
12	Chrisinger and King [38]	M	2 (App)	-	(1) Phone-based GPS in App	(Audio and image from phone App)	(2) Empatica 4 (f = 4 Hz)	Sensors	(a) Normalize the Electrodermal activity (EDA) from E4 by subtracting the minimum; (b) center (subtracting the mean) and scaled (dividing by the standard deviation of the centered data) the EDA data; (c) Use an algorithm to remove the noise from EDA data.	-	Over spatial units (set 5-m grid cells along the walk path to use Getis-Ord Gi* local statistic and kernel density to detect cluster).
13	Dessimond, et al. [39]	L	2	(1) Canarin (Air pollution)	(2) Tablet-based GPS	-	-	Remote server/ Sensor	-	-	Over spatial units (based on specific Locations) and over time (hour).
14	Do, et al. [40]	L	3	(1) PM monitor for Air pollution (15 s sampling rate); (2) Temperature logger.	(3) GPS receiver (5 s sampling rate) and a Wi-Fi hotspot	-	-	Cloud server/ Sensors (If Wi-Fi connectivity was unavailable)	(a) Assigned all missing PM measurements as “−9999”; (b) Clean GPS data by the distance between two points; e.g., assign distance > 50 as “NaN”; (c) Co-locate the PM data with air monitoring site to adjust the data.	Time interpolation (from 15 s to 5 s)	Over Time and spatial units
15	Doherty and Oh [41]	M	3+1 (App)	-	(1) GPS receiver (every 1 s)	(2) Accelerometer from Electrocardiogram	(2) Electrocardiogram (25 measurements per second); (3) Glucose monitoring (every 10 s).	Phone App and remote server	(a) A rule-based algorithm to detect human activity from GPS data; (b) Average glucose readings every 5-min.	A Web-based retrospective data analysis software.	-
16	Donaire-Gonzalez, et al. [42]	M	5+1 (App)	(1) Black carbon monitor MicroAeth; (2)UFP monitor DiSCmini; (every 1 s).	(3)Phone-based GPS in App; (4) GPS ^8^ receiver (every 10 s).	(3) Phone-based Acceleromer in App; (5) Accelerometer (every 10 s).	-	Phone App and cloud server	Phone App used to process the data by algorithm.	Phone App (every 10 s)	-
17	Doryab, et al. [43]	H	2 (App)	(1) Phone App to record social activity (1 sample per 10 min).	(1) Phone-based GPS in App	-	(2) A Fitbit Flex (sleep at 1 sample per min, and steps at 1 sample per 5 min).	Sensor and server	(a) Develope a feature extraction component (FEC) to extract features; (b) Handle Missing Values, e.g., removed a participant if 20% data were missing.	-	Over time (all day, night, morning, afternoon, weekdays and weekend).
18	El Aarbaoui and Chaix [44]	H	4	(1) Personal Dosimeter (every second)	(2) GPS receiver	(3) Accelerometer (5 s epochs)	(4) BioPatch BHM 3	Sensors	-	Over 5-min and 1-min windows with the coefficient of variation.	Over spatial units (based on different contexts).
19	Engelniederhammer, et al. [45]	M	3	(1) Infrared motion sensor (f = 10 Hz)	(2) GPS receiver	-	(3) A wristband developed by Bodymonitor (EDA data with f = 10 Hz).	The Infrared data was transmitted to wristband and stored in sensor	(a) A classification algorithm to detect emotion based on EDA data; (b) Reduced the data to binary information and use the logit model to deal with them.	-	-
20	Huck, et al. [46]	L	3+1 (App)	(1) NO_2_ sensor (f = 1 Hz)	(2) Phone based GPS (f = 1 Hz)	-	(3) Airflow Sensor (f = 1 Hz)	Phone App	Phone App	Phone App	Over spatial units
21	Johnston, et al. [47]	L	2+1 (App)	(1) PM_2.5_ monitor (every second); (2)Phone:Temperature, humidity.	(2) Phone-based GPS (every second)	-	-	Phone App	Phone App	Phone App	Over spatial units and time (hour) and individuals.
22	Kanjo, et al. [48]	H	2+1 (App)	(1) Phone: Noise sensor; (2) Microsoft band: Air pressure and Light.	(1) Phone-based GPS	-	(2) Microsoft band (App-based self-report)	Phone App	(a) The first and the last 30 s were cut; (b) Remove abnormal ones by lagged Poincare plots.	Phone App	-
23	Kim, et al. [49]	M	3 (App)	-	(1) Phone-based GPS in App	(2) Accelerometer	(3) Empatica 4 (f = 4 Hz)	Sensors	(a) Use Butterworth low-pass filter with a cut-off frequency of 4 Hz to remove noise from accelerometer data; (b) Use time interpolation to unify the frequency of GPS data (f = 1 Hz).	Average value over subsegment (61 in total)	-
24	Kou, et al. [50]	M	2	(1) Sound sensor: sound level (minute-by-minute).	(2) Phone-based GPS (at a resolution of 1 m or 3 s)	-	-	Sensors	Classify activity time into day, evening and night; Classify activity companion into “alone” and “with others”; Classify activity type into “work and study”, “personal affairs”, “housework”, “shopping” and “recreation”.	-	Over time (use a logarithmic function to aggregate the fluctuating sound levels over a period of time).
25	Laeremans, et al. [51]	H	2	(1) MicroAeth: expsoure to black carbon (on a five-minute basis).	-	(2) Accelerometer from SenseWear	(2) SenseWear armband (on a one-minute basis).	Sensors	(a) Use SenseWear professional software to extract feature; (b) Choose bouts of at least 10 consecutive minutes with an intensity ≥3 METs ^9^; (c) Raw black carbon (BC) data were smoothened with the Optimized Noise-reduction Algorithm (ONA) ^10^.	-	Over individuals (amount, percentage, mean and standard deviation).
26	Ma, et al. [52]	H	2	(1) Sound Meters: Noise level (1 min).	(2) Phone-based GPS	-	-	Sensors	(a) Classify the activities into categories; (b) Use a-weighted equivalent sound pressure level to estimate the average noise exposure.	-	Over time and spatial units (Average the parameters based on the time and duration for each category of activity or travel mode on a weekday and weekend day).
27	Ma, et al. [53]	M	2	(1) Portable Air monitor (1 s)	(2) Phone-based GPS (1 s)	-	-	Sensors	-	-	Over spatial units and individuals (sum of the per second exposure for each person).
28	Millar, et al. [54]	M	3 (App)	(1) Camera (participants’ view)	(2) Phone-based GPS From App (f = 1 Hz)	(head activity from camera)	(2) (Empatica 4 (f = 4 Hz)	Sensors	(a) Use weighted moving average with a 60-s moving window to compute smoothed speed from GPS App; (b) Weights were re-normalized and they summed to 1; (c) Extract Skin conductance responses(SCRs) from EDA by Ledalab; (d) Use a moving window of 20 s to identify deviations of SCR; (e) Standardized the SCR to reduce differences between participants.	Time interpolation (f = 4 Hz)	Over spatial units and time (a web-based mapping system to visualize high-resolution spatiotemporal data).
29	Novak, et al. [55]	M	2	(1) PM measuring unit (1 min) A reference instrument: GRIMM ^11^	-	-	(2) Smart Activity tracker: Garmin Vivosmart 3 (in minute)	Sensors	-	-	Over time (5-min averages).
30	Ojha, et al. [56]	M	3	(1) Sensor backpack monitoring Sound and dust (f = 0.4 Hz), Temperature, illuminance (f = 1 Hz).	(2) GPS receiver (f = 1 Hz)	-	(3) Empatica 4 (f = 4 Hz)	Sensors	(a) Remove unusable EDA data; (b) Filter EDA data to remove artifacts; (c) Smooth data by Stationary Wavelet Transform; (d)Time window marking; (e) Extract Skin conductance responses(SCRs) from EDA by Ledalab; (f) Data labeling: “normal” and “aroused”.	Apply Time interpolation (f = 1 Hz) to environment data, and keep health data at original frequency (f = 4 Hz).	Over individuals (the mean physiological response across all participants and normalized between 0 and 1).
31	Rabinovitch, et al. [57]	M	3	(1) Aerosol, nephelometer: fine PM concentrations; (2) Temperature sensor (10 s intervals).	(3) GPS receiver (10 s intervals)	-	(An electronic monitor of school-time albuterol use: total number)	Sensors	(a) Use an algorithm to classify the types of microenvironment; (b) Use a normalization factor to correct measurement.	-	Over time (both mean and 1-min maximum) and spatial units (based on contexts).
32	Resch, et al. [58]	M	4+1 (App)	(1) GoPro camera (First-person video camera).	(2) Phone-based GPS	-	(3) Empatica 4, Zephyr, (4) Bioharness (ECG, HRV). (eDiary App)	Sensors/Phone App	(a) Filter data by a low-pass filter (f = 0.5 Hz) and a high-pass filter (f = 0.05 Hz); (b) Use a rule-based algorithm to detect pattern of stress.	-	Over spatial units (aggregated to raster cells and use Getis–Ord Gi hotspot analysis).
33	Roe, et al. [59]	M	4 (App)	(1) Noise sensors, (2) Air monitor.	(3) GPS from App (f = 1/60 Hz) (Phone App)	(3) Accelerometer from App (f = 60 Hz)	(3) Huawei watch (Photoplethysmogram with f = 100 Hz).	Sensors (noise and air monitor) and phone App(smart watch)	t-test to determine any significant difference between parameters.	-	-
34	Runkle, et al. [60]	H	2	(1) Temperature sensor (5-min)	(2) GPS from Garmin smartwatch		(2) Garmin smartwatch (1-min)	Sensors	(a) Categorize temperature data into “extreme heat” and “moderate heat”; (b) Caculate the average of maximum heart rate over a 5-min interval.	Reduction (5-min)	-
35	Rybarczyk, et al. [61]	M	3 (App)	(1) GoPro Hero (images about road)	(2) Tablet-based GPS from App	(3) Accelerometer from Garmin	(3) Garmin VívoSmart (1-s)	Sensors	(a) Removed GPS errors and missing data in GIS manaully and by the “remove duplicate” records tool in ArcGIS; (b) Normalize physiological data by using inverse distance weighting (IDW) in ArcGIS to create a smoothed raster surface.	Interpolation (spatially joined the interpolated values to track point layer to produce a completed and normalized database).	Over spatial units (Average based on spatial configuration of the environment).
36	Shoval, et al. [62]	M	2+1 (App)	-	(1) Phone-based GPS from App (1 min)	-	(2) Empatica4 (f = 16 Hz) and phone (Phone App based location-triggered and time-triggered surveys).	Sensors and phone	Calculate z-scores for each measurement to normalize Skin Conductance Level (SCL).	Frequency Reduction (mean SCL z-scores over 1 min)	Over individuals and spatial units (based on 20 m× 20 m cellular network).
37	Steinle, et al. [63]	L	2	(1) Dylos 1700 for measuring PM concentrations (1 min)	(2) GPS receiver (every 10 s)	-	-	Sensors	(a) Classify Microenvironment into six types	Match data by the Feature Manipulation Engine software (Safe software Inc., 2014) at every full minute.	Over time (hours) and spatial units (Microenvironment types) and individuals.
38	West, et al. [64]	L	2	(1) Dylos 1700	(2) GPS receiver (every 10 s)	-	-	Sensors	-	Average in timeframes (1 min)	Over time (each 30 min period) and spatial units (based on each 50 m grid square) and individuals.
39	Zhang, et al. [65]	H	4 (App)	(1) Noise sensors (one-minute intervals); (2) Air sensors (1 s) connected a Phone App; (3) A mobile signal detection device.	(4) Phone-based GPS (f =1 Hz)	-	-	Sensors and Phone App	Use A-weighted equivalent sound pressure level to calculate the sound exposure.	-	Over time (Average value of A sound level for a certain period of time).

^1^ Q indicates the result of quality assessment. ^2^ N indicates the number of equipment used in each application. ^3^ Ledalab software is a free MATLAB-based tool to process Electrodermal activity (EDA) and identify skin conductance response (SCR). ^4^ WIMU Studio is an in-house developed open-source software (https://github.com/introlab/openwimu (accessed on 8 November 2021)). ^5^ MVPA, moderate-to-vigorous physical activity, assessed by accelerometry. ^6^ Personal Activity Location Measurement System (PALMS) is an encrypted web application to simultaneously processes time-stamped accelerometer and GPS data. ^7^ “+1” indicates the additional assistance from smartphone and phone-based application. ^8^ To validate the accuracy of GPS and accelerometry from the phone, participants also carried a GPS tracker and an accelerometer, attached to the same belt of the smartphone. ^9^ METs stands for Metabolic Equivalent of Task, used to express exercise intensity. ^10^ Optimized Noise-reduction Algorithm (ONA), developed by the United States’ Environmental Protection Agency. ^11^ GRIMM (Durag Group, Hamburg, Germany) Model 11-A (1.109) Aerosol Spectrometer (GRIMM) is a reference instrument for PM measurements with five-minute resolution.

**Table 4 sensors-21-07693-t004:** How to improve the quality of integration.

Preparation	
Recruitment of participants	Follow the requests of ethics approval to protect data [36,37,48,51,60].
	Larger samples in terms of gender, age and socioeconomic background [35,49]; random selection will be suggested [59]. Measure participants’ demographics [38]; Subpopulations or vulnerable groups such as children and older people who have more complex demands need attentions [41].
Fieldwork design	Focus on the narrower geographic area to find similar discrepancies between individual assessments of the same feature [38], but diverse assumed conditions [45,60]; More cities and locations [42,45,51,58]; Culture, language and ethnicity [34]; Season and weather [34]. Confounders controlling will be suggested [44,52].
Time-series measurement	Ranging from a few days to several months [29] to collect longer-term data may reveal additional information [43]; Repeat in the different seasons [33,51].
Multiple sensor selection	
Objective	Synthesize the elements related to the objectives; Study from literature and make current applications as a reference;
Choose sensors	These should be considered: Choose commercial sensors or self-developed sensors; The cost-effectiveness and availability, including the price, sources and data protection [37,49]; The functionality and comfortability, including appearance, size, weight, and carrying [30,42]; The total weight, number and size of sensor package and the processing of carry sensors package [63].
Test the accuracy of sensors	Test the generality and performance in different areas and situations [43,52,53]; Consider using additional sensors to improve the prediction accuracy [42]; Co-locating the sensors with monitoring sites to test or adjust the data [40]. All devices should be synchronized using the timestamp and internal clocks before the study [36].
Data collection	
User’s operation	Decrease human factors: All the participants, research assistants and technicians should be trained before real implementation [41,50,51] to follow the steps of operation of sensors, since incorrectly wearing may lead to inaccuracy [30]. Ensure high quality data: Follow the guidelines of usage [33]. Check, charge and calibrate the sensors daily to prevent sensor failure [31,33,44,64,65].
Avoid	Decrease non-human factors include environmental noises and technical noises: Signal weakness/loss [30,64]; Low battery [63]; Incomplete measures and underestimations [34];
Data load	Store data in the sensors [29,30,31]. Store data in the smartphone [20,21,22]. Transmit data to the central server [40,41]. Data fusion can be done during data collection [48].
Data integration	
Data processing	Clean the noise and unwanted signals: Utilize specific software to automate the matching process and extract features [27,34,63]; Utilize algorithm to remove malfunctioned, negative or zero values [51,56]; Utilize a “filter” to cut off data below the threshold [30,31,49].
Normalization	Average data within the time window [30,49]. Normalize all data between 0 and 1 [35]. Use statistics/algorithm to normalize the data [38,57].
Frequency unification	Pair the frequency: Time interpolation [49,56]; Frequency reduction [29,62].
Aggregation	Aggregate data for statistics and visualization Aggregate over multiple individuals [35,62]; Aggregate over space [38,62]; Aggregate over time [43,50].
New development	Develop an integrated system (e.g., smartphone, web-platform, software) to automatically process sensor data, store and visualize due to its portability and accessibility [41,42].

## Data Availability

Not applicable.

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
