# Peer review of "Assessing the Current Integration of Multiple Personalised Wearable Sensors for Environment and Health Monitoring"

_sensors, 2021, doi:10.3390/s21227693_

Round 1

Reviewer 1 Report

This paper presents a systematic review of the data integration for environment and health monitoring. The review protocol follows establish guidelines, with valid search strategy, selection criteria, and quality assessment method. The reviewed papers cover a reasonable span of research period. The authors identified two main challenges: sensor selection and data synchronization, and data fusion and storage. Several strategies were proposed to improve the quality of data integration in the target context. Although many of the issues raised in the paper is already well-known for any kind of data integration problem (e.g., data synchronization, resampling, cost-effectiveness), this paper did generated some new insights specific to the target context (e.g., the spatial aspect of the data integration).

There are two additional issues that are relevant to a data integration in general, but are not mentioned in this paper, include the accessibility of raw data and the unification of terminology. For example, consumer health trackers such as Fitbit and Apple Watch do not support the export of raw sensor signals and only processed data can be exported. That could significantly affect the subsequent data analysis. I was wondering whether any of the reviewed papers touched this issue. In addition, different manufacturers may adopt their own 'original' naming system that does not comply with the standard terminology. For example, in sleep tracking, the total sleep duration is called 'total sleep time (TST)' by medical devices, 'minutes asleep' by Fitbit, 'sleep time' by Neuroon. Did any of the reviewed paper mention this issue? 

Many of the strategies listed in Table 4 are informative but lack technical depth. Especially in the data integration section, each of the subitem can be further expanded. What signal processing techniques (e.g., what kind of filters) have been used for noise removal and which one(s) demonstrated the best performance? What is the best window size for moving average? How does up sampling (aka time interpolation) and down sampling (aka frequency reduction) affect the final analysis outcome? What are the algorithms available for different kinds of data aggregation and what are their most suited application scenarios? At a higher level, do these proposed strategies apply to all health aspects? It's likely that data integration techniques should be adapted to the target health metric. 

Author Response

Thank you for your valuable feedback on our manuscript. We really appreciate your thoughtful comments and suggestions. We have thoroughly gone through your comments and think they have considerably improved our paper. We have made the following revisions according to your suggestions.

1. Comments: There are two additional issues that are relevant to a data integration in general, but are not mentioned in this paper, include the accessibility of raw data and the unification of terminology. For example, consumer health trackers such as Fitbit and Apple Watch do not support the export of raw sensor signals and only processed data can be exported. That could significantly affect the subsequent data analysis. I was wondering whether any of the reviewed papers touched this issue. In addition, different manufacturers may adopt their own 'original' naming system that does not comply with the standard terminology. For example, in sleep tracking, the total sleep duration is called 'total sleep time (TST)' by medical devices, 'minutes asleep' by Fitbit, 'sleep time' by Neuroon. Did any of the reviewed paper mention this issue?

Responses:  That is true. The accessibility of raw data should be considered at the beginning of choosing sensors. We now report this issue in section 3.3.3 (Line 250-254). For example, Shoval, et al. mentioned that they choose E4 because the raw data can be easier exported, and Birenboim, et al. mentioned that the data from the MS band cannot be exported directly without the assistance of a third-party Android mobile application. But the data quality of Fitbit, and the issue with another commercial band, such as Garman, was not reported in reviewed papers.

     Secondly, since the application of wearable sensors is still at an earlier stage, so the terminology is not standardized to some degree. We discuss this issue in the section of discussion (Line 414-420). First, to compare the data from different sensors, it is essential to comply with the standardised terminology and nomenclature according to the official regulations [55]. For example, Butt, et al. assessed the duration of sleep by “total sleep time (TST)”, “the number of awakenings”, and “time taken to fall asleep” measured by medical devices, while Doryab, et al. assess sleep by the average length of “asleep” or “awake” measured by Fitbit. The inconsistency of terminology may increase the difficulties to repeat the experiment with different sensors, but we did not find any of the reviewed papers mentioning this issue, we recommend describing the features of sensors and data by unified criteria at the end of the discussion.

       Besides, we also suggest describing the features of the sensor in a unified format (Line 420-423), for example, the frequency of the sensor in a different format, such as “frequency “, “every 1s”, “1 recording 1 second “,”1-min epochs”.

2. Comments: Many of the strategies listed in Table 4 are informative but lack technical depth. Especially in the data integration section, each of the subitem can be further expanded. What signal processing techniques (e.g., what kind of filters) have been used for noise removal and which one(s) demonstrated the best performance? What is the best window size for moving average? How does up sampling (aka time interpolation) and down sampling (aka frequency reduction) affect the final analysis outcome? What are the algorithms available for different kinds of data aggregation and what are their most suited application scenarios? At a higher level, do these proposed strategies apply to all health aspects? It's likely that data integration techniques should be adapted to the target health metric.

Responses:  Table 4 is a conceptual framework for integrating sensors and data, strategies mentioned here are for general data integration issues, we added more technical details of processing data in Table 3, so readers can go back to check the information if they need a specific strategy.

We also discussed the following questions in section 3.4.

- What signal processing techniques (e.g., what kind of filters) have been used for noise removal and which one(s) demonstrated the best performance?

Addressed (Line 292-298). Butterworth filter performed well on removing higher or lower frequency variations. For example, Resch, et al. filtered Galvanic skin response (GSR) by a low-pass filter at 0.5 Hz and a high-pass filter at 0.05 Hz; Boissy, et al. [32] filtered accelerometer data by the low-pass filter at 5Hz and high-pass filter at 1Hz; and Kim, et al. filter accelerometer data by a low-pass filter with cut-off frequency at 4Hz.

- What is the best window size for moving average?

Addressed (Line 316-324). The determination of window size for moving average is dependent on the features of the data. For health data, we recognize that the determination of window size for moving average should also follow the health metric since human responses have different latency times. For example, 5min windows for measuring short-term Heart Rate Variability (HRV), and 1min windows for assessing HRV dynamics that may be masked within 5-min windows. Thus, the decision-making window size for moving average is a critical process.

- How does up sampling (aka time interpolation) and down sampling (aka frequency reduction) affect the final analysis outcome?

Addressed (Line 305-311). We discussed the merits and shortcomings of upsampling and downsampling. For example, time interpolation could keep the higher resolution of datasets but may increase the computation, while downsampling can save the effort of computation, but may lead to the loss of resolution.

- What are the algorithms available for different kinds of data aggregation and what are their most suited application scenarios?

Addressed (Line 340-345). As for the algorithm, we mainly discuss the aggregation of the spatial units. This is because, for the majority of reviewed papers, the aggregation by time and individuals can be realized through the calculation of average, mean value within individuals and timeframe, but the aggregation by spatial units needs specific software and algorithms.

- At a higher level, do these proposed strategies apply to all health aspects? It's likely that data integration techniques should be adapted to the target health metric.

Response: The general recommendations made in table 4 are necessarily generic guidelines. For specific health outcomes or environmental exposures, modifications to these guidelines would be expected (Line 354-356).

Reviewer 2 Report

Review for Assess the current integration of multiple personalised weara- 2 ble sensors for environment and health monitoring

Information not extracted accurtately from the articles included in this systematic review, e.g., 1. informed consent was obtained in both references #27 & 28 which are the first two entries in the list of included articles. (Benita et al, and Benita & Tuncer).

  1. Donaire-Gonzales Ref# in the main paper, report an average age of 61 for their participants whereas Table 2 in the appendix indicate no reporting of age.
  2. Shimazaki and Katsu’s (Ref#62) research was performed in japan and not Finland.

This reviewer has selected these four items randomly and have not verified the accuracy of other extracted information and strongly recommend authors doing so. Accordingly please modify the text.

The inclusion/exclusion criteria, specifically excluding studies that were performed indoor does not sound logical. It would have been a reasonable criterion if the goal of the review was to verify the effect of outdoor pollution/stressor on the health aspects but as is its logic needs further explanation. Also, for the purpose of this paper it would be probably ok not to include non-Engligh articles but in general language is not an acceptable exclusion criterion.

On line#40, please define PM2.5

On line#100-101, “tracked individual passively” What does this mean? If by passive you mean using passive sensors, then at least some of the sensors reported in the included articles are not passive such as Fitbit. Many of the wearable wrist monitors, if not all, use LED light for HR sensing and that makes them active sensor. Similarly, EDA sensors are not passive. Either way, please define this terminology.

Please word out EPHPP

Line#150-151, authors comment on the geographical distribution of using multiple sensors. A s only English language papers were included in this review, is it possible that the geographical location/distribution of these studies are affected by that? Please discuss. The same comment applies to “The current applications are unequally distributed in the world” on line# 165-166

The article needs occasional rewording.

Author Response

Thank you for your valuable feedback on our manuscript titled “Assessing personal exposure to urban greenery using wearable camera and machine learning”. We really appreciate your thoughtful comments and suggestions. We have thoroughly gone through your comments and think they are of great importance in improving our paper and inspiring our further research. We have made the following revisions according to your suggestions.

1. Comments: Information not extracted accurtately from the articles included in this systematic review, e.g., 1. informed consent was obtained in both references #27 & 28 which are the first two entries in the list of included articles. (Benita et al, and Benita & Tuncer).

  • Donaire-Gonzales Ref# in the main paper, report an average age of 61 for their participants whereas Table 2 in the appendix indicate no reporting of age.
  • Shimazaki and Katsu’s (Ref#62) research was performed in japan and not Finland.

This reviewer has selected these four items randomly and have not verified the accuracy of other extracted information and strongly recommend authors doing so. Accordingly please modify the text.

Responses:  Thank you. We have undertaken a thorough review of the papers included in this review. We have revised the information in Appendix Tables and all tables in the main paper where necessary, we also added more details.

2. Comments: The inclusion/exclusion criteria, specifically excluding studies that were performed indoor does not sound logical. It would have been a reasonable criterion if the goal of the review was to verify the effect of outdoor pollution/stressor on the health aspects but as is its logic needs further explanation. Also, for the purpose of this paper it would be probably ok not to include non-Engligh articles but in general language is not an acceptable exclusion criterion.

Responses: We revised the introduction (Line 36-38) and refined the study objectives, which now focus on the outdoor environment (Line 57-61). The goal of this paper is to assess the integration of multiple wearable sensors in urban environments when people are conducting outdoor activities.

Based on that, we exclude Shimazaki and Katsu’s paper, since it monitors people’s walking at 4 km/h on a treadmill.

While we agree studies should not be excluded based on language, the authors do not have enough competency in other languages to effectively review non-English language papers. We note that English is the dominant language of science communication, and it is reasonable to limit ourselves to English papers (Line 85-86).

3. Comments: On line#40, please define PM2.5

Responses:  Addressed (Line 40-41).

4. Comments: On line#100-101, “tracked individual passively” What does this mean? If by passive you mean using passive sensors, then at least some of the sensors reported in the included articles are not passive such as Fitbit. Many of the wearable wrist monitors, if not all, use LED light for HR sensing and that makes them active sensor. Similarly, EDA sensors are not passive. Either way, please define this terminology.

Responses: Addressed. We have revised the term and replaced it with the phrase “collected data automatically via sensing technology” (Line 97-98).

5. Comments: Please word out EPHPP

Responses: Addressed (Line117-118). EPHPP is the abbreviation of Evaluation of Public Health Practice Projects.

6. Comments: Line#150-151, authors comment on the geographical distribution of using multiple sensors. As only English language papers were included in this review, is it possible that the geographical location/distribution of these studies are affected by that? Please discuss. The same comment applies to “The current applications are unequally distributed in the world” on line# 165-166

Responses:  Thank you, we admitted that the language barriers may lead to the unequal distribution of these studies and mentioned in the section of 3.1(Line 147-149).

7.Comments: The article needs occasional rewording.

Responses:  We have thoroughly reviewed the text and made changes to improve the readability

Round 2

Reviewer 1 Report

I appreciate the authors' effort in addressing the review comments. The manuscript reads well and I'd be happy for it to get published as is. 

Reviewer 2 Report

The reviewer is satisfied with the revision.